# Prebiotic Effect of Maitake Extract on a Probiotic Consortium and Its Action after Microbial Fermentation on Colorectal Cell Lines

**DOI:** 10.3390/foods10112536

**Published:** 2021-10-21

**Authors:** Alessandra De Giani, Federica Bovio, Matilde Emma Forcella, Marina Lasagni, Paola Fusi, Patrizia Di Gennaro

**Affiliations:** 1Department of Biotechnology and Biosciences, University of Milano-Bicocca, 20126 Milano, Italy; alessandra.degiani@unimib.it (A.D.G.); f.bovio@campus.unimib.it (F.B.); matilde.forcella@unimib.it (M.E.F.); paola.fusi@unimib.it (P.F.); 2Department of Earth and Environmental Sciences, University of Milano-Bicocca, 20126 Milano, Italy; marina.lasagni@unimib.it

**Keywords:** prebiotics, mushroom, Maitake, probiotics, functional food, nutraceuticals

## Abstract

Maitake (*Grifola frondosa*) is a medicinal mushroom known for its peculiar biological activities due to the presence of functional components, including dietary fibers and glucans, that can improve human health through the modulation of the gut microbiota. In this paper, a Maitake ethanol/water extract was prepared and characterized through enzymatic and chemical assays. The prebiotic potential of the extract was evaluated by the growth of some probiotic strains and of a selected probiotic consortium. The results revealed the prebiotic properties due to the stimulation of the growth of the probiotic strains, also in consortium, leading to the production of SCFAs, including lactic, succinic, and valeric acid analyzed via GC-MSD. Then, their beneficials effect were employed in evaluating the vitality of three different healthy and tumoral colorectal cell lines (CCD841, CACO-2, and HT-29) and the viability rescue after co-exposure to different stressor agents and the probiotic consortium secondary metabolites. These metabolites exerted positive effects on colorectal cell lines, in particular in protection from reactive oxygen species.

## 1. Introduction

The nutritional and medicinal effects of mushrooms are recognized worldwide, in particular in Asian and northern and Central American countries. Indeed, over 270 fungal species are identified for their biological activities, such as anti-inflammatory, antimicrobial, and antioxidant properties [1]. Nevertheless, an important role has always been recognized in the preservation of the healthy state of the gastrointestinal tract [2]. Accordingly, nowadays, mushrooms are used not only in the pharmaceutical industry but also in the nutraceutical and cosmeceutical industries [1]. This is due to their high protein and low-fat contents, as well as to the presence of several vitamins, and minerals [1], but most of all, it is due to the presence of important functional components such as dietary fibers, chitin, and glucans [3]. In particular, glucans are polysaccharides composed of chemically heterogeneous glucose molecules, classified in α- or β-glucans depending on the glycosidic linkage [4]. β-glucans are known for their beneficial effects in lowering blood pressure, reducing glycemia, and acting as antitumoral and antioxidant agents [5]. Furthermore, they could be used by the intestinal microbiota as prebiotics [2], defined as “non-digestible dietary food ingredients that, when passing through the colon, will benefit the host by selectively stimulating the growth and/or activity of one or a limited number of beneficial bacteria in situ” [6]. Therefore, they are currently the most sought after functional food [2].

Among the mushrooms with health booster properties, there is *Grifola frondosa* (knows as Maitake, Hen of the wood, or Signorina mushroom) [7]. It belongs to the Polyporaceae family [8], and it is characterized by caps with a smoky brown color [7]. Generally, it grows in temperate regions, such as Japan, Europe, northeastern states of America, and subtropical regions at high elevations [7]. Regarding the chemical composition, Maitake is characterized by 3.8% water-soluble polysaccharides; among them, 13.2% correspond to (1 → 3, 1 → 6)-β-d-glucans [7]. Typically, the water-soluble polysaccharides can be divided into two subpopulations based on the molecular weight, i.e., 722.7 kDa and 19.6 kDa [9].

Other components are starch, oligofructose, fructooligosaccharides (FOS), lactulose, galactomannan, polydextrose, and dextrin [10]. Because of these characteristics, researchers are trying to combine the possible beneficial effects of the mushroom with the helpful action of probiotics (i.e., live microorganisms that confer a health benefit to the host when administered in adequate amounts [11]). Generally, this type of fermentation is applied to several food ingredients because probiotics can enhance the nutritional value of food [5] and support positive health effects, such as immune modulation and the maintenance of a state of eubiosis in the context of the gut microbiota composition. Furthermore, mushrooms themselves could improve the antioxidant condition through the modulation of the gut microbiota [10]. As a consequence, an important role in the status of intestinal mucosal epithelial cells could be played by the released bacterial secondary metabolites generated by mushrooms’ fermentation, in particular, short-chain fatty acids (SCFAs) [12], including acetate, propionate, butyrate, and valerate [13]. In the presence of a medium-rich fiber diet, SCFA concentration in the intestine is around 10 mmol/L, so the intestinal epithelial cells are constantly exposed to these metabolites, mediating the crosstalk between the microbiota and the host [14]. It is known that a fiber-rich diet could also prevent the development of colorectal cancer (CRC). Interestingly, the SCFA butyrate could especially act as an antitumoral agent through the modulation of several transduction pathways, including the cellular proliferation pathway [12]. For example, butyrate inhibited the proliferation of a colorectal tumoral cell line, decreasing the presence of the phosphorylated extracellular-regulated kinase 1/2 (p-ERK 1/2), which is classified as a survival signal [14].

In this study, we analyzed a commercially available Maitake (*Grifola frondosa*) dried extract, characterized for its β-glucan content. Based on the specifics, we tested the prebiotic property on several probiotic strains, comprising both *Lactobacillus* and *Bifidobacterium* genera. Then, we studied a powerful probiotic consortium that, in the presence of Maitake preparation, released SCFAs. The fermentation products were then tested for their effects on the vitality of both healthy and tumoral colorectal cell lines. Finally, the rescue of the viability of a cell line after induced stresses was evaluated.

## 2. Materials and Methods

### 2.1. Preparation of the Maitake Extract 

The Maitake (*Grifola frondosa* Dicks. Gray) extract was obtained from Amita HC Italia S.r.l. (Solaro, Milan, Italy). The original mushrooms came from China, and they were wild at the time of the manual collection (from August to November). Then, the sporophorum part was selected for the extraction of polysaccharides.

The commercial Maitake dried extract was prepared as follows. The Maitake sporophorums were first ground and weighted. Then, to obtain an extract enriched in polysaccharides, the obtained material was resuspended in a solution of ethanol:water (20:80 ratio), and the separation was allowed. After overnight incubation, the precipitate was collected and then dried at 50 °C to eliminate the ethanol. Finally, the obtained extract was blended and sieved to create a brownish fine powder characterized by a particle size lower than 180 µm.

### 2.2. Characterization of the Maitake Extract

#### 2.2.1. Determination of Starch Molecules Content

The starch molecules content was estimated through the Megazyme kit K-TSHK (Megazyme Inc., Chicago, IL, USA) as described by the manufacturer’s instructions.

A starting weight of 100 mg of Maitake extract powder was used for the measurement. Then, 0.2 mL of aqueous ethanol (80% *v*/*v*) was added and then stirred using a magnetic stirrer. An amount of 2 mL of 2 M KOH was added, and the solution was stirred for 20 min in an ice-water bath. Then, 8 mL of 1.2 M sodium acetate buffer (pH 3.8) was added, together with 0.1 mL of thermostable α-amylase (from Megazyme kit) and 0.1 mL of amyloglucosidase (20 U, from Megazyme kit). All the contents were stirred and then incubated at 50 °C for 30 min. The obtained solution was centrifuged at 3000 rpm for 10 min, and 0.1 mL of the supernatant was analyzed.

#### 2.2.2. Determination of α- and β-Glucans Content

α- and β-glucans content was estimated through the Megazyme kit K-YBGL (Megazyme Inc., Chicago, IL, USA) as described by the manufacturer’s instruction. A starting weight of 100 mg of Maitake extract powder was used for the measurement. Then, 2 mL of ice-cold 2 M KOH was added and then stirred using a magnetic stirrer at 4 °C for 20 min. In total, 1.2 M sodium acetate buffer was added, and then amyloglucosidase (1630 U/mL) plus invertase (500 U/mL) (200 μL) (from Megazyme kit) was added. All the contents were mixed and then incubated at 40 °C for 30 min. The obtained solution was centrifuged at 2000 rpm for 10 min and 0.1 mL of the supernatant was analyzed for the glucose presence. For the total glucan measurement, 100 mg of Maitake extract powder was used. An amount of 2 mL of ice-cold 12 M sulfuric acid was added and then stirred. The tubes were then placed at 4 °C for 2 h in agitation. Then, 10 mL of water was added to each sample, which was placed in a hot water bath (100 °C) for 2 h. After cooling at room temperature, 6 mL of 10 M KOH was added, and the content was mixed well. Then, the volume was adjusted to 100 mL with 200 mM sodium acetate buffer at pH 5. In total, 100 µL of the sample was incubated with 100 µL of a mixture of exo-1,3-β-glucanase (20 U/mL) plus β-glucosidase (4 U/mL) at 40 °C for 60 min, and the glucose was determined with GOPOD reagent (all of the reagents used were in the Megazyme kit). Absorbance was measured at 510 nm. A 0.1 mL aliquot of 1 mg/mL glucose standard solution was incubated in triplicate with GOPOD reagent; 0.1 mL of acetate buffer (200 mM, pH 5) was also incubated with 3.0 mL of GOPOD reagent as a blank sample. Finally, the β-glucan content was determined by subtracting the α-glucan content from the total glucan content.

The calculations were made through Mega-Calc sheet (Megazyme Inc., Chicago, IL, USA).

#### 2.2.3. Determination of Polyphenols Content

The total phenolic content of the Maitake extract was measured using the Folin–Ciocalteau phenol assay previously described [15]. A standard solution of Gallic Acid (GA) ranging from 0 to 100 µg/mL was used for the calibration. The GA solutions were prepared in 80% methanol (Sigma, Milano, Italy), and the absorbance values were measured at 765 nm. For the quantification, 0.5 mL of Folin-Ciocalteau phenol reagent (1:10 dilution) and 1 mL of distilled water were added to 100 μL of mushroom sample. The solutions were mixed and incubated at room temperature for 1 min. Then, 1.5 mL of 20% sodium carbonate (Na_2_CO_3_) solution was added to the sample and mixed. After the incubation for 120 min, absorbance was recorded at 765 nm against the blank sample. Results were expressed as mg of Gallic Acid Equivalent (GAE)/g of Maitake powder. All the measurements were made in triplicate.

#### 2.2.4. Determination of Protein Content

The protein content of Maitake extract was determined according to the Bradford method [16]. A calibration curve using bovine serum albumin as a standard was performed to determine the protein concentration of the extract. 

#### 2.2.5. Determination of Fructans and Reducing Sugars Content

The quantification of fructans in the Maitake extract powder was developed using the fructan assay procedure kit Megazyme [17] in accordance with the manufacturer’s instructions. The fructans concentration was calculated considering the fructose, glucose, and sucrose contents in the mushroom extract before and after the hydrolysis with fructanase. The samples were treated with a specific sucrase/maltase enzyme to completely hydrolyze saccharides to d-glucose and d-fructose. The reference values of the samples were determined by a direct analysis of d-glucose plus d-fructose using the hexokinase/phosphoglucose isomerase/glucose 6-phosphate dehydrogenase analytical procedure. The amount of NADPH formed in this reaction is stechiometric with the amount of d-glucose plus d-fructose. NADPH formation is measured by the increase in absorbance at 340 nm. The fructan content of the samples was determined after hydrolyzation to d-fructose and d-glucose by endo- and exo-inulinases, and then d-fructose and d-glucose content was measured as described above. The fructan content was determined by subtracting absorbance values of the reference from those of the sample. Beforehand, each enzymatic assay sample was heated for 30 min at 50 °C to ensure sample complete dissolution. 

### 2.3. Bacterial Strains and Culture Conditions

The bacterial strains used in this study are reported in Table 1. The strains provided by a private collection of the company Roelmi HPC (Origgio, Italy) were previously selected and characterized for the probiotic features by [18]. The probiotic strains were routinely grown on MRS broth (Conda Lab, Madrid, Spain) supplemented with 0.03% l-cysteine (Sigma-Aldrich, Milano, Italy) for 48 h, at 37 °C, under anaerobic conditions using an Anaerocult GasPack System (Merck, Darmstadt, Germany). A modified MRS medium (mMRS), as described by [19], without glucose and supplemented with 0.03% l-cysteine, was used for the growth trials. The pH of the medium was adjusted to 6.8 before sterilization (121 °C for 20 min). Maitake extract was added to mMRS at the concentration of 2% *w*/*v* as a carbon source. 

### 2.4. Growth Experiment with Single Probiotic Strains on Maitake Preparation

The probiotic bacteria described in Table 1 were pre-inoculated in MRS broth for 48 h, at 37 °C in anaerobic conditions before the setup of the prebiotic experiment. Maitake extract powder was added to mMRS and then sterilized together before inoculation, to a final concentration of 2% *w*/*v*.

In total, 1 mL of sterile mMRS or sterile mMRS + Maitake preparation was added to every well of a sterile multi-well (24 wells, SPL Lifesciences, Pocheon-si, Korea). Then, each well was inoculated with a proper volume of each probiotic pre-inoculum (around 20 μL), to achieve a final optical density (OD) at 600 nm of 0.1. Subsequently, the plates were capped and incubated in an anaerobic jar at 37 °C for 48 h. At the end of the experiment, the OD at 600 nm was measured.

### 2.5. Growth Experiment with Mixed Probiotic Strains as Consortium on Maitake Preparation

The probiotic bacteria described in Table 1 were pre-inoculated in MRS broth for 48 h at 37 °C in anaerobic conditions before the setup of the experiment. Maitake extract powder was added to mMRS and then sterilized together before inoculation, to a final concentration of 2% *w*/*v*.

The consortium was prepared in a sterile tube, mixing the selected probiotic in order to achieve an OD at 600 nm of 0.1. After the homogenization, the proper volume of the consortium (around 200 μL) was inoculated in sterile tubes (BD, Milano, Italia) with a final volume of 10 mL. Each tube contained only mMRS as a control or mMRS plus Maitake extract. After the inoculum, the tubes were placed in an anaerobic jar and then incubated for 48 h at 37 °C. The growth was evaluated as optical density at 600 nm.

### 2.6. Extraction and Characterization of Probiotics Secondary Metabolites

#### 2.6.1. Extraction of the Metabolites

The possible produced short-chain fatty acids (SCFAs) after Maitake fermentation were extracted from inoculated and uninoculated tubes at the end of the fermentation using ethyl-acetate (anhydrous, 99.8%, Sigma-Aldrich, Milano, Italy). The tubes were centrifuged at 7000 rpm for 10 min at room temperature to separate the pellet from the supernatant. In a glass tube (Colaver, Vimodrone, MI, Italy), 5 mL of the supernatant was acidified up to pH 2 with HCl 6 M. Then, 5 mL of ethyl-acetate was added, followed by 20 min of strong manual agitation. The obtained suspension was centrifuged at 4000 rpm for 20 min, and the organic phase was withdrawn and conserved in a new glass tube. Another 5 mL of ethyl-acetate were added to the remaining broth, followed by 5 min of strong manual agitation. The suspension was again centrifuged, and the organic phase collected and pooled with the one obtained in the first extraction.

#### 2.6.2. Analysis of the Extracted Metabolites

The extracted organic phase was submitted to derivatization with BSTFA (Sigma-Aldrich, Milano, Italy) before the gas chromatography (GC) injection. The samples and BSTFA were mixed in a ratio of 3:1 and the reaction took place at 60 °C for 20 min. After the temperature cooled down, SCFAs were analyzed with a GC-MSD instrument, using a Technologies 6890 N Network GC System, interfaced with a 5973 Network Mass Selective Detector (MSD) (Agilent Technologies, Santa Clara, CA, USA). A ZB-5MS capillary column was used (5% diphenyl-95% dimethylpolysiloxane 60 m × 0.25 mm, 0.25 μm; Alltech, Lexington, KY, USA). Analyses were developed in the splitless injection mode, using helium at 99.99% as carrier gas (Sapio, Bergamo, Italy). The program for the oven was set at 65 °C for 2 min, then 5 °C min^−1^ to 110 °C, then 12 °C min^−1^ to 260 °C, holding this temperature for 10 min. Electron impact ionization spectra were obtained at 70 eV, with recording of specific mass spectra at 73, 75, 117, 129, 132, 145, 159, 171, 173, 187, 201, 215, 229, 243, and 257 *m*/*z*. All the analyses were carried out in triplicate.

The registered mass spectra were compared with those of the library of National Institute of Standards and Technology (NIST) of the instrument.

### 2.7. Maintenance and Growth of Cell Lines for In Vitro Tests

The healthy mucosa cell line CCD841 (ATCC^®^ CRL-1790™) and the colon cancer cell line CACO-2 (ATCC^®^ HTB-37™) were grown in EMEM medium supplemented with heat-inactivated 10% FBS, 2 mM l-glutamine, 1% nonessential amino acids, 100 U/mL of penicillin, and 100 µg/mL of streptomycin and maintained at 37 °C in a humidified 5% CO_2_ incubator; the colon cancer cell line HT-29 (ATCC^®^ HTB-38™) was grown in DMEM medium supplemented with heat-inactivated 10% FBS, 2 mM l-glutamine, 100 U/mL of penicillin, and 100 µg/mL of streptomycin and maintained at 37 °C in a humidified 5% CO_2_ incubator. 

ATCC^®^ cell lines were validated by short tandem repeat profiles that were generated by the simultaneous amplification of multiple short tandem repeat loci and amelogenin (for gender identification).

All the reagents for cell cultures were supplied by EuroClone (EuroClone S.p.A, Pero, MI, Italy).

### 2.8. Cell Viability Assay and Test with Stress Agents

Cells were seeded in 96-well microtiter plates at a density of 1 × 10^4^ cells/well and incubated for 24 h. 

To evaluate the effect of the extracts on CCD841, CACO-2 and HT-29 viability, the cells were treated for 24 h with 0.25, 0.5, and 1 mg/mL for each extract. 

HT-29 cell line was treated with stressor compounds, H_2_O_2_ (0–8 mM), or SDS (0–0.1%), for 24 h in order to determine the concentration responsible for an about 50% reduction in cell viability. To evaluate a potential role of the extract in cell viability rescue, cells were pre-treated for 1 h with 0.25, 0.5, and 1 mg/mL for each extract, and then 1 mM H_2_O_2_ or 0.0075% SDS was added to the cells with a 24 h endpoint. Following the treatment, cell viability was assessed using an in vitro MTT-based toxicology assay kit (Sigma, St. Louis, MO, USA): the medium was replaced with a complete medium without phenol red, and 10 µL of 5 mg/mL MTT (3-(4,5-dimethylthiazol-2)-2,5-diphenyltetrazolium bromide) solution was added to each well; after 4 h of incubation, formed formazan crystals were solubilized with 10% Triton-X-100 in acidic isopropanol (0.1 N HCl) and absorbance was measured at 570 nm using a micro plate reader. The results were expressed as mean values ± ES of at least three independent experiments.

### 2.9. Statistical Analysis

Regarding the growth of probiotics, all the experiments were performed in triplicate and results were presented as mean values ± standard deviation. The statistical relevance of the results was assessed by a *t*-Student’s test. The significance was defined as *p*-value < 0.01 or *p*-value < 0.05.

Regarding the experiments with the intestinal cell lines, all the experiments were performed in triplicate. The results were shown as the mean value of vitality % ± standard error. Statistical differences were calculated using Dunnett’s multiple comparisons test: * *p*-value < 0.05, ** *p*-value < 0.01, *** *p*-value < 0.001. 

## 3. Results

### 3.1. Characterization of the Principal Components of Maitake Extract

The determination of the main components of the Maitake extract was evaluated with enzymatic assays using Megazyme methods, as described in the Materials and Methods Section. The major components of the mushroom preparation are starch (around 50% *w*/*w*), and glucans (around 25% *w*/*w*, comprising α- and β-glucans). Interestingly, the presence of α-glucans is very low with respect to the β-ones, which are dominant in the extract (around 6.2% vs. 18.8%, respectively). Furthermore, Maitake extract was also analyzed for its content of polyphenols and protein by chemical reactions. Each extract preparation contained at least 1.9% *w*/*w* of polyphenols and 0.02% *w*/*w* of proteins. Other components are sugars; indeed, the extract is characterized by 1.2% *w*/*w* of fructans and 3.6% *w*/*w* of free reducing sugars. The contents of glucans, fructans, free reducing sugars, starch, polyphenols and protein are listed in Table 2.

### 3.2. Prebiotic Potential of Maitake Preparation on Lactobacillus and Bifidobacterium Strains

The possible prebiotic capability of Maitake extract at a concentration of 2% *w*/*v* was evaluated through in vitro growth assays employing *Lactobacillus* and *Bifidobacterium* strains, originally isolated from the human colon [18]. The initial OD_600nm_ of each culture was 0.1; then, it was recorded at the end of the experiment, after 48 h of anaerobic fermentation (Figure 1). The medium containing all components except the extract was used as a control. As shown in Figure 1, the mushroom preparation could be considered as a prebiotic substrate for the probiotic bacteria, because all the tested strains were able to ferment it and the difference between the growth in the control condition and the growth on the prebiotic was statistically significant (*p*-value < 0.01). Among the *Lactobacillus* strains, *L. fermentum*, *L. rhamnosus*, and *L. reuteri* showed the most positive responses (*p*-value < 0.05), while, regarding the Bifidobacteria members, all the strains showed an important growth increase in Maitake extract, in particular *B. longum*/*infantis* and *B. lactis* (*p*-value < 0.01). The results highlight the prebiotic characteristic of the mushroom formulation.

### 3.3. Growth of the Probiotic Consortium on Maitake Preparation

To enhance the possible beneficial effects for the human host attributable both to probiotics and prebiotics, a possible combination of the bacteria was studied. All the probiotic strains previously used were inoculated at OD_600nm_ equal to 0.1, making sure to have a homogeneous suspension of all the considered bacteria. The growth capacity was evaluated on the same conditions used for the experiment with single strains, i.e., control medium and Maitake extract 2% *w*/*v*. After 48 h of anaerobic growth at 37 °C, the optical density was measured (Figure 2). Interestingly, the final OD on the mushroom preparation had a mean value of 3.7 ± 0.21. The prebiotic potential of Maitake preparation is also confirmed in this kind of experiment, because of the very significant difference between the growth on the sole medium and the one on the medium added with Maitake extract (*p*-value < 0.01). 

### 3.4. Extraction and Characterization of Probiotic Consortium Secondary Metabolites

The production of potentially beneficial secondary metabolites derived from the fermentation of complex carbohydrates by probiotics is well documented. They are recognized as short-chain fatty acids (SCFAs) and as branched-chain fatty acids (BCFAs) [20].

To investigate which compounds are produced by the probiotic combination after the Maitake extract fermentation, a liquid–liquid extraction with ethyl-acetate of the broth culture after 48 h of anaerobic fermentation was initially conducted. Then, a gas chromatography analysis was performed, and the chromatograms were interpreted using mass spectrometry. Each peak was compared to the example present in the NIST library. 

In comparison with the control condition, samples deriving from fermented Maitake extract presented several additional peaks (Figure 3). The highest peak, at a retention time of 8.2 min, refers to lactic acid. The second most important peak is at 12.2 min of retention time, and it is assigned to valeric acid. The third highest peak refers to succinic acid (retention time of 13.6 min). Other detected molecules were the SCFA butyrate (retention time of 10.8 min) and hydrocinnamic acid (retention time of 16.8 min), which is not a bacterial secondary metabolite, but a Maitake component [21] probably released due to probiotic digestion, and cinnamic acid (retention time of 19.9 min).

### 3.5. Differential Effects of Extracted Secondary Metabolites on Cell Viability

To understand the possible role of the previously characterized extracted secondary metabolites, viability assays on both healthy and tumoral cell lines were developed.

The healthy mucosa cell line CCD841 was not affected by the treatment with the metabolites (Figure 4), with the exception of the fermented Maitake extract, responsible for a 20% reduction in cell viability at the highest concentration analyzed (Figure 4B).

Regarding the two colorectal cancer cell lines, no variation in vitality was observed in the HT-29 cell line (Figure 4), while the CACO-2 cell line behaved quite differently. In fact, in this cell line, the viability showed an about 20% reduction at a Maitake extract concentration of 1 mg/mL (Figure 4A), as well as after exposure to fermented probiotic basal medium extract (Figure 4C). The treatment with the Maitake–probiotic fermented extract showed a significant dose-dependent decrease in cell viability already detectable at the lowest dose (Figure 4B). 

### 3.6. Protective Effect of Maitake-Probiotic Fermented Extract on HT-29 Cell Line

Since the HT-29 cell line represents a valuable and complementary tool for the study of food digestion and the effect of food components on the gut [22], the evaluation of a potential protective effect exerted by the extracts in stress conditions was conducted on this cell line. 

Firstly, the cell survival rate in the presence of several stressors at different concentrations was investigated (data not shown). In the presence of either 1 mM H_2_O_2_ or 0.0075% SDS, cell viability compared to control was found to be 42.97 ± 5.49% and 56.13 ± 3.07%, respectively. Subsequently, co-treatment with the stressor compounds and the different extracts at the established concentrations demonstrated how only the Maitake–probiotic fermented extract protects the cells from H_2_O_2_ challenge (Figure 5A), but not from SDS stress (Figure 5B). Furthermore, the protective effect was only shown in the low–middle doses (Figure 5). 

## 4. Discussion

Extracts deriving from vegetables naturally contain polysaccharides that are characterized by different chemical compositions [23]. These molecules can be extracted and employed as nutraceuticals, leading to several beneficial effects for the host and its gut microbiota. In particular, mushrooms are known for their high content of polysaccharides with antioxidant and immunomodulatory potential [23]. Among the most interesting polysaccharides, there are glucans, which are a group of chemically heterogeneous glucose molecules, classified as α- or β-glucans based on the glycosidic linkage [4]. 

The Maitake extract preparation tested in this study contains around 25% of total glucans, in line with the study of [4]. Interestingly, in the Maitake water extract analyzed in this study, the presence of α-glucans is very low with respect to the β- ones (around 6.2% vs. 18.8%, respectively), and this is in accordance with the literature. Indeed, high levels of α-glucans are not naturally present in mushrooms [4]. Moreover, β-glucans are not cleaved by mammalian digestive enzymes [24], so they can reach the intestine and be used as carbon and energy sources, as well as metabolized to other compounds by the resident microbiota. Therefore, to be considered a prebiotic, the Maitake preparation also has to benefit the host by the selective stimulation of the growth of a limited number of bacteria in the colon [6], i.e., the probiotics. For this reason, we tested the potential prebiotic effect on different *Lactobacillus* and *Bifidobacterium* strains previously characterized for their probiotic features [18]. All the tested strains were able to ferment the Maitake preparation at a concentration of 2% *w*/*v*, reaching a statistically significant growth difference with respect to the control. The best growth results were obtained with *L. fermentum* LF, *L. reuteri* LR, and *B. animalis* spp. *lactis* BL. The maximum growth values on Maitake preparation are similar to the ones of known prebiotics, i.e., inulin or fructooligosaccharides (FOS). For example, the growth values of the same probiotic strains on commercial FOS with a degree of polymerization comprised between 3 and 5 at a concentration of 2% *w*/*v* [25] are similar to the ones obtained in this study, after 48 h of Maitake preparation fermentation at the same concentration, that are around an optical density at 600 nm of 4. 

Therefore, the Maitake preparation could be considered a prebiotic molecule, in line with the literature data. Indeed, Ref. [26] studied the possible prebiotic effect of polysaccharides extracted from 53 different wild-growing mushrooms on *Lactobacillus acidophilus* (used as reference strain) and *Lactobacillus rhamnosus*. The extracted fungal molecules could stimulate the Lactobacilli growth more than commercially available prebiotics, such as inulin and FOS from chicory. Additionally, Bifidobacteria could benefit from the presence of β-glucans in the growth medium, as reported by [27] in the reconstruction of the catabolic pathway of *B. longum* subspecies *infantis* strain.

Generally, probiotics are provided as a unique individual strain. However, multi-strain formulations could boost the beneficial effects for the host compared to single-strain products [28]. For this reason, some probiotic strains (five different *Lactobacillus* and three different Bifidobacteria) were combined and tested, again confirming the prebiotic potential of the Maitake preparation. 

Furthermore, the metabolites derived from the fermentation of the mushroom preparation by the consortium were characterized for their possible impact on the host intestinal cells. This is a new concept reported in the literature as “metabiotics” [27], i.e., signaling molecules with a determined chemical structure that can optimize host-specific physiological functions and regulate metabolic and behavior reactions [29]. Indeed, the production of potentially beneficial secondary metabolites deriving from probiotics’ fermentation on different carbon sources as complex carbohydrates, is well known [20]. These released molecules are commonly recognized as short-chain fatty acids (SCFAs) and include acetate, propionate, and butyrate, which comprise more than 95% of the total SCFA pool. Less abundant released metabolites are branched-chain fatty acids (BCFAs), such as isobutyrate, 2-methylbutyrate, isovalerate, lactate, and succinate (intermediates of the propionate), which can also have several effects on the host [20]. From the consortium broth culture, lactic acid, valeric acid and butyrate were obtained after liquid extraction. 

In a gut microbiota ecosystem, the bacteria that catabolize specific molecules to simpler ones, which could be fermented and result in acidic molecules, such as acetate or lactate, are classified as primary degraders. Then, the secondary fermenters could use the secondary metabolites from the primary degraders to produce other end products, such as butyrate [30]. Among the community, Bifidobacteria are interesting because they promote butyrate release by other community members due to the production of acetate and lactate through the characteristic Bifid shunt [30]. Nevertheless, the microbiota could ferment proteins, which represent 0.02% *w*/*w* of the Maitake preparation, resulting in BCFA production [20], and it could liberate and modify bioactive polyphenols, as the detected cinnamic and hydrocinnamic acids are. These molecules create a physiological response in the host despite the low availability due to the metabolism of the original complex molecules by the gut microbiota [20]. 

In general, in the literature, the beneficial effects for the host of all these secondary metabolites are largely discussed. Indeed, lactic acid promotes the balance of intestinal pH [28], while butyrate also modulates the immune system, as well as the strength of the epithelial barrier, and could be protective against colorectal cancer [30]. Therefore, creating microbiota through a beneficial composition could be important for the host’s health. However, considering the intestinal system, the first barrier of the human body in contact with the lumen is composed of epithelial cells. 

Therefore, the extracted and characterized metabolites after the probiotic fermentation of the Maitake preparation were tested on different healthy and tumoral colorectal cell lines. Among the extracts, only the Maitake–probiotic fermented extract caused a significative dose-dependent reduction in the CACO-2 cell line, with the HT-29 and CCD841 not being affected at 0.25 and 0.5 mg/mL doses, while at the highest concentration used, the healthy one also showed a slight reduction. These results are in accordance with the paper by [14], in which the authors demonstrated the inhibitory effect of butyrate on cancerous cell lines. The different responses of HT-29 and CACO-2 to the Maitake–probiotic fermented extract treatment could be due to a different mutational status on the downstream EGFR target BRAF. In fact, HT-29 cells present a hyperactivating mutation in BRAF, responsible for constitutive ERK1/2 phosphorylation, as shown in the paper by [31]. Consequently, the human colon adenocarcinoma cell line, HT-29, was selected for further studies. Indeed, the cell line is a good model for studies on food digestion and bioavailability because the cells can form a monolayer characterized by tight junctions and the typical apical brush border, representative of mature intestinal cells [22]. Thus, a possible protective effect due to the probiotic secondary metabolites in different stress conditions was evaluated. As the results showed (Figure 5), the protection is principally against the H_2_O_2_ challenge, because HT-29 vitality recovers by up to about 80%. This is important, because the formation of reactive oxygen species (ROS) is common in human organs, as a result of the oxidative processes; however, they are precursors of systemic cells and tissue damage [21]. Generally, the human body has an endogenous defense system against these free radicals; however, it can be supported and potentiated by supplemented antioxidants [21]. 

In conclusion, this study demonstrates the prebiotic properties of Maitake extract and its bioactive compounds. These molecules are used as carbon and energy sources by different bacteria strains belonging to *Lactobacillus* and *Bifidobacterium* genera. The combination of the eight probiotic strains and their fermentation of the mushroom preparation leads to the production of beneficial secondary metabolites that have positive effects on colorectal cell lines. In particular, they promote the recovery of cell viability after the stress induced by ROS species. 

Therefore, in the midst of the pandemic caused by COVID-19, eating healthy foods that reinforce our gut microbiota and have intrinsically functional properties could be a way to prevent and improve our defenses [23]. The optimization of the composition of the probiotic consortium and the choice of the bioactive natural extracts to obtain new synbiotic formulations with a powerful antioxidant potential on the intestinal cells and a beneficial modulatory potential on host gut microbiota could be valuable alternative supports. 

## 5. Conclusions

In conclusion, the results revealed the prebiotic properties of Maitake extract due to the stimulation of the growth of the probiotic strains, also in consortium, leading to the production of SCFAs, including lactic, succinic, and valeric acid. These metabolites exerted positive effects on colorectal cell lines, in particular protecting from reactive oxygen species. These data indicate that prebiotics from Maitake could be suitable for use in food applications and could be combined with probiotics in synbiotic formulations.

## Figures and Tables

**Figure 1 foods-10-02536-f001:**
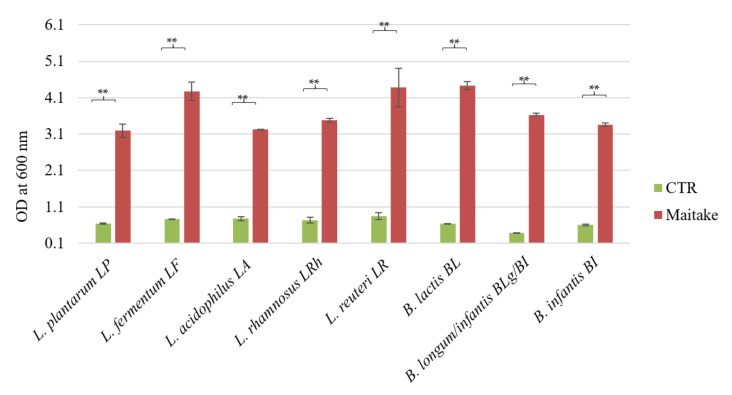
Prebiotic potential of Maitake extract on *Lactobacillus* and *Bifidobacterium* strains. The Figure represents the growth of the selected *Lactobacillus* and *Bifidobacterium* strains in presence of CTR medium and Maitake extract at a concentration of 2% *w*/*v*. Values are represented as mean value of OD at 600 nm ± standard deviation. Statistical differences were calculated using *t*-Student’s test: ** *p*-value < 0.01.

**Figure 2 foods-10-02536-f002:**
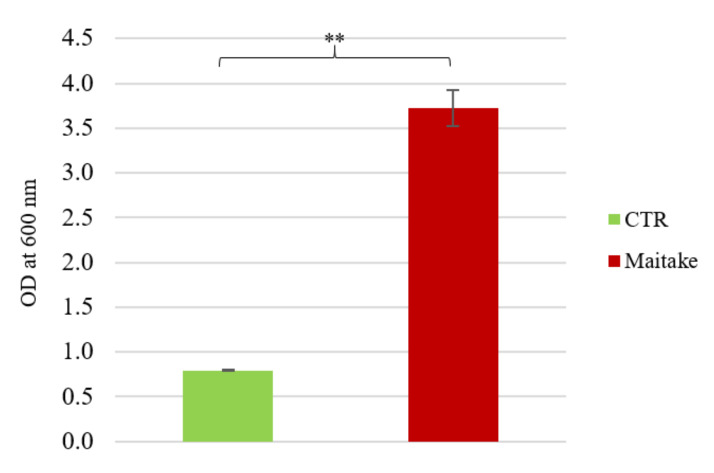
Prebiotic potential of Maitake extract on probiotic consortium. The figure represents the growth of consortium of the selected probiotic strains in presence of CTR medium and Maitake extract at a concentration of 2% *w*/*v*. Values are represented as mean value of OD at 600 nm ± standard deviation. Statistical differences were calculated using *t*-Student’s test: ** *p*-value < 0.01.

**Figure 3 foods-10-02536-f003:**
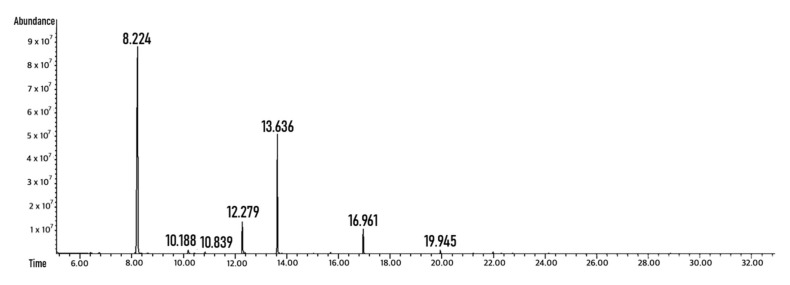
Probiotic consortium secondary metabolites analysis by GC-MSD after growth on Maitake extract. The figure represents the chromatogram obtained after GC-MSD analysis. The probiotic consortium was incubated for 48 h on Maitake extract at a concentration of 2% *w*/*v*. The cultural broth was extracted and then analyzed in GC-MSD. The intermediate metabolites identified are reported in the graph.

**Figure 4 foods-10-02536-f004:**
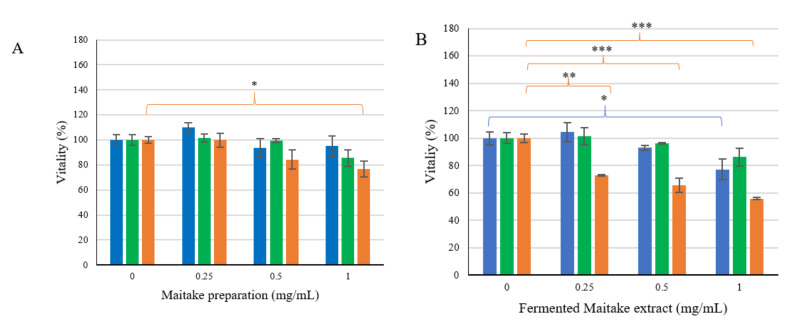
Differential effects of extracted secondary metabolites on cell viability. The figure represents the viability of the three different cell lines in presence of Maitake extract (**A**), fermented Maitake extract (**B**), and fermented control medium extract (**C**). Values are represented as mean value of vitality % ± standard error. Statistical differences were calculated using Dunnett’s multiple comparisons test: * *p*-value < 0.05, ** *p*-value < 0.01, *** *p*-value < 0.001.

**Figure 5 foods-10-02536-f005:**
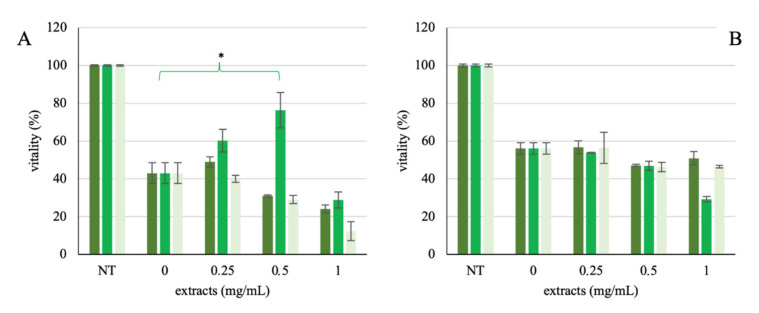
Vitality rescue of HT-29 cell line co-treated with stressor compounds and different extracts. The figure represents the viability rescue of the HT-29 cell line stressed with H_2_O_2_ (**A**) or SDS (**B**) in presence of Maitake preparation (
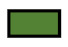
), fermented Maitake water extract (
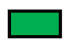
), and fermented basal medium extract (
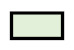
). Values are represented as mean value of vitality % ± standard error. Statistical differences were calculated using Dunnett’s multiple comparisons test: * *p*-value < 0.05.

**Table 1 foods-10-02536-t001:** Bacterial strains used in this study.

Strain	Source
*Lactobacillus acidophilus* LMG P-29512 (FORMERLY DSM 24936)	Human
*Lactobacillus fermentum* DSM 25176	Human
*Lactobacillus plantarum* DSM 24937	Human
*Lactobacillus reuteri* DSM 25175	Human
*Lactobacillus rhamnosus* LMG P-29513 (FORMERLY DSM 25568)	Human
*Bifidobacterium animalis* SPP. *lactis* LMG P-29510 (FORMERLY DSM 25566)	Human
*Bifidobacterium longum* SPP. *longum* DSM 25174	Human
*Bifidobacterium longum* SPP. *infantis* LMG P-29639	Human

**Table 2 foods-10-02536-t002:** Composition of Maitake extract used in this study.

	Maitake Extract (%)
Starch	49.5
Total glucans *alpha*-Glucans *beta*-Glucans	25.0 6.2 18.8
Proteins	0.02
Polyphenols	1.9
Fructans	1.2
Reducing sugars	3.6

## Data Availability

All data generated during this study are included in this article.

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
