# Peer review of "Prebiotic Effect of Maitake Extract on a Probiotic Consortium and Its Action after Microbial Fermentation on Colorectal Cell Lines"

_foods, 2021, doi:10.3390/foods10112536_

Round 1

Reviewer 1 Report

This is a vey interesting manuscript, containing an useful information bot from the scientific level and also for its possible future applications

The results obtained, revealed the prebiotic properties of Maitake extract due to the stimulation of the growth of the probiotic strains, also in consortium, leading to the production of SCFAs, including lactic, succinic, and valeric acid

These metabolites exerted positive effects on colorectal cell lines, in particular protecting from reactive oxygen species.

These data indicate that prebiotics from Maitake could be suitable for use in food applications and could be combined with probiotics in symbiotic formulations.

Author Response

The reference has been added in the Introduction.

The Revised Manuscript with the marked revisions required by the two Reviewers is in attachment.

Reviewer 2 Report

The manuscript by De Giani et al. reported that Maitake extract works as prebiotics and influences the viability of colorectal cell lines. The authors examined the prebiotic effect of Maitake extract on Lactobacillus and Bifidobacterium strains as well as the probiotic consortium. Secondary metabolites were analyzed in the probiotic consortium, presumably induced by the Maitake extract. Fermented Maitake extract inhibited CACO-2 colorectal cancer cells, and protect from cell death in HT-29 cells after H2O2 treatment. The idea of this study is interesting; however, the manuscript is not well-organized. The authors need to revise the structure of the manuscript. Specific comments are shown below.

  1. le 1 and Table 2 can not be found in the main text.
  2. The Discussion and the result are the same. It seems a simple mistake. Please check them.
  3. The protective effect on HT-29 cells from H2O2 treatment was shown. Why is it not observed in SDS treatment?
  4. What are the differences between normal and fermented Maitake extracts? Have the authors examined what ingredients were changed?
  5. Why did Maitake extract differently affect HT-29 and CACO-2 colorectal cancer cell lines? It would be better to show the possible explanation in the discussion.
  6. On page 1, lines 24-29 should be deleted.

Author Response

Response to the Reviewer 2

The manuscript by De Giani et al. reported that Maitake extract works as prebiotics and influences the viability of colorectal cell lines. The authors examined the prebiotic effect of Maitake extract on Lactobacillus and Bifidobacterium strains as well as the probiotic consortium. Secondary metabolites were analyzed in the probiotic consortium, presumably induced by the Maitake extract. Fermented Maitake extract inhibited CACO-2 colorectal cancer cells, and protect from cell death in HT-29 cells after H2O2 treatment. The idea of this study is interesting; however, the manuscript is not well-organized. The authors need to revise the structure of the manuscript. Specific comments are shown below.

1. Table 1 and Table 2 can not be found in the main text.

Reply: we are sorry for this, but it depends not from us. Table 1 and 2 are forgotten from members of the journal staff in reporting the text of the original manuscript submitted in the format of the journal. Now the tables have been added. 

2. The Discussion and the result are the same. It seems a simple mistake. Please check them.

Reply: we are sorry for this, but it depends not from us. The mistake derives from members of the journal staff in reporting the text of the original manuscript submitted in the format of the journal. Now the Discussion have been added.

3. The protective effect on HT-29 cells from H2O2 treatment was shown. Why is it not observed in SDS treatment?

Reply: We thanks the reviewer for the question. We assumed that our Maitake-probiotic fermented extract exerts a protective role on H2O2 challenge, probably due to a positive effect on antioxidant defence mechanisms; while the SDS stress, damaging cell membranes, is more difficult to revert. Further studies will be focused on the investigation of the Maitake-probiotic fermented extract’s antioxidant properties, with a deeply analysis on the antioxidant defences involved. 

4. What are the differences between normal and fermented Maitake extracts? Have the authors examined what ingredients were changed?

Reply: We thanks the reviewer for the question. We examined the differences between the non-fermented Maitake extract and the probiotic-fermented Maitake extract through GC-MS analysis. In the basal medium+Maitake extract only succinic acid and stearyl citrate (formerly 1,2,3-Propanetricarboxylic acid, 2-hydroxy-, octadecyl ester) are present. While, in the fermented Maitake extract, there are SCFAs, and BCFAs (as shown by the reported figure in the main text of the paper). In particular, in the fermented extract, the succinic acid presents a higher peak with respect to the not-fermented sample. 

5. Why did Maitake extract differently affect HT-29 and CACO-2 colorectal cancer cell lines? It would be better to show the possible explanation in the discussion.

Reply: We thanks the reviewer for the question. In the paper Zeng et al 2017 (Zeng H, Taussig DP, Cheng WH, Johnson LK, Hakkak R. Butyrate Inhibits Cancerous HCT116 Colon Cell Proliferation but to a Lesser Extent in Noncancerous NCM460 Colon Cells. Nutrients. 2017 Jan 1;9(1):25. doi: 10.3390/nu9010025. PMID: 28045428; PMCID: PMC5295069.) it has been shown the inhibitory effect of butyrate on P-ERK1/2 only in cancerous colon cells, while it exerts a positive effect on healthy cell line. The different response of HT-29 and CACO-2 following Maitake extracts treatment could be due to a different mutational status on the EGFR pathway, responsible for a constitutive ERK1/2 phosphorylation. In fact, HT-29 cells present a hyperactivating mutation in BRAF, upstream to ERK1/2, as shown in the paper Bovio et al 2020 (Bovio F, Epistolio S, Mozzi A, Monti E, Fusi P, Forcella M, Frattini M. Role of NEU3 Overexpression in the Prediction of Efficacy of EGFR-Targeted Therapies in Colon Cancer Cell Lines. Int J Mol Sci. 2020 Nov 20;21(22):8805. doi: 10.3390/ijms21228805. PMID: 33233823; PMCID: PMC7699864.).

The possible explanation has been added to the discussion as suggested by the Reviewer.

Round 2

Reviewer 2 Report

All the concerns were addressed in the current state of the manuscript.